# SNARE Protein Snc1 Is Essential for Vesicle Trafficking, Membrane Fusion and Protein Secretion in Fungi

**DOI:** 10.3390/cells12111547

**Published:** 2023-06-05

**Authors:** Muhammad Adnan, Waqar Islam, Abdul Waheed, Quaid Hussain, Ling Shen, Juan Wang, Gang Liu

**Affiliations:** 1Shenzhen Key Laboratory of Microbial Genetic Engineering, College of Life Sciences and Oceanography, Shenzhen University, Shenzhen 518060, China; alvi.adnan@yahoo.com (M.A.); waheed90539@gmail.com (A.W.); wangjuan@szu.edu.cn (J.W.); 2College of Physics and Optoelectronic Engineering, Shenzhen University, Shenzhen 518060, China; 3Xinjiang Key Laboratory of Desert Plant Roots Ecology and Vegetation Restoration, Xinjiang Institute of Ecology and Geography, Chinese Academy of Sciences, Urumqi 830011, China; ddoapsial@yahoo.com; 4Shenzhen Key Laboratory of Marine Bioresource and Eco-Environmental Science, College of Life Sciences and Oceanography, Shenzhen University, Shenzhen 518060, China; quaid_hussain@yahoo.com; 5School of Life Science, Jiangsu Normal University, Xuzhou 221116, China; lingshen@jsnu.edu.cn

**Keywords:** SNC1, SNARE proteins, protein trafficking, vesicle fusion, SNARE complex, protein secretion

## Abstract

Fungi are an important group of microorganisms that play crucial roles in a variety of ecological and biotechnological processes. Fungi depend on intracellular protein trafficking, which involves moving proteins from their site of synthesis to the final destination within or outside the cell. The soluble N-ethylmaleimide-sensitive factor attachment protein receptors (SNARE) proteins are vital components of vesicle trafficking and membrane fusion, ultimately leading to the release of cargos to the target destination. The v-SNARE (vesicle-associated SNARE) Snc1 is responsible for anterograde and retrograde vesicle trafficking between the plasma membrane (PM) and Golgi. It allows for the fusion of exocytic vesicles to the PM and the subsequent recycling of Golgi-localized proteins back to the Golgi via three distinct and parallel recycling pathways. This recycling process requires several components, including a phospholipid flippase (Drs2-Cdc50), an F-box protein (Rcy1), a sorting nexin (Snx4-Atg20), a retromer submit, and the COPI coat complex. Snc1 interacts with exocytic SNAREs (Sso1/2, Sec9) and the exocytic complex to complete the process of exocytosis. It also interacts with endocytic SNAREs (Tlg1 and Tlg2) during endocytic trafficking. Snc1 has been extensively investigated in fungi and has been found to play crucial roles in various aspects of intracellular protein trafficking. When Snc1 is overexpressed alone or in combination with some key secretory components, it results in enhanced protein production. This article will cover the role of Snc1 in the anterograde and retrograde trafficking of fungi and its interactions with other proteins for efficient cellular transportation.

## 1. Introduction

Fungi are a versatile group of organisms that have widespread applications in industry, medicine, and agriculture [1]. Filamentous fungi are composed of elongated cells with multiple nuclei, enabling a complex and dynamic protein transportation system [2]. The regulation of protein transport is critical for the growth and survival of filamentous fungi, necessitating a comprehensive understanding of this mechanism. Protein synthesis initiates in the cytoplasm and the synthesized proteins are targeted to specific organelles or compartments within the cell [3]. This transportation occurs via the secretory pathway, which involves several steps. The proteins are first synthesized by ribosomes and transferred to the endoplasmic reticulum (ER). The ER carries out protein folding and modifications before being forwarded to the Golgi apparatus via transport vesicles. In the Golgi apparatus, the proteins undergo additional modifications and are finally sorted into vesicles for transport to their ultimate destinations [4]. The vesicles are transported along the cytoskeleton towards the target organelle or plasma membrane (PM) [3]. Finally, the membrane of the vesicle and that of the target membrane fuse with each other, which results in the discharge of the cargo proteins into the organelle or extracellular space [4].

Membrane fusion at the target compartment is facilitated by a group of specialized proteins known as SNAREs (Soluble N-ethylmaleimide-sensitive factor attachment protein receptors). These proteins play a critical role in ensuring the effective and coordinated fusion of the two membranes [5]. The orderly fusion of membranes requires four SNARE proteins, which form a complex via the interaction of their alpha helices. This brings the two membranes in close proximity, with the concomitant expulsion of water molecules between them. The complex formed by this trans-SNARE interaction is known as SNAREpin, while the SNARE proteins involved in this process are classified as either v-SNAREs (which are associated with the vesicle membrane) or t-SNAREs (which are associated with the target membrane) [6]. The membrane fusion is completed by recruiting an R-SNARE and a set of three Q-SNAREs, named after their specific target residues, namely arginine (R) and glutamine (Q), respectively [6,7,8]. SNARE proteins are present on a wide range of membranes, including the PM, Golgi membranes, endosomes, vacuoles, ER, and their derived vesicles [6].

Snc1 is a v-SNARE protein that plays a crucial role during the final stages of protein secretion. Specifically, it plays an indispensable role in the process of vesicle fusion with Spitzenkörper and PM [9]. The exocyst complex of yeast (*Saccharomyces cerevisiae*) comprises the Rab-GTPases Sec4, Rho3, and CDC42. This complex is responsible for the tethering of secretory vesicles to the PM, which is necessary for their final fusion, and this process is equally mediated by SNARE proteins [10]. The process of vesicle fusion at the PM is initiated by the formation of a binary complex between the Sso1/2 and Sec9 t-SNAREs, which later bind with Snc1 (Figure 1) [11]. Interestingly, in *Trichoderma reesei*, Snc1 was shown to interact with Sso2 in the apical regions, while SSO1 interacts with Snc1 in the subapical regions [9]. This localization of SNAREs at the PM suggests that there are specific routes for lateral secretion. In summary, the coordinated action of SNARE proteins, including Snc1, is essential for the regulated secretion of proteins and other molecules from cells. The specificity of the interaction between SNARE proteins and their regulators, such as the exocyst complex, provides a level of control that is essential for proper cellular function.

Generally, interactions between Snc1 and other SNARE proteins are regulated by several accessory proteins such as SM proteins (Sec1/Munc18) and Ypt1 Rab GTPase (Table 1) [12]. The “zippering” of the trans-SNARE complex brings the vesicular and target membranes in close proximity, leading to their fusion [13]. The SNARE complex then dissociates on the target membrane, aided by N-ethyl maleimide-sensitive factor (NSF) and Sec18, and enables SNARE recycling [14]. Although the mechanism of Snc1-mediated vesicle fusion in filamentous fungi is similar to that in *S. cerevisiae*, some notable differences exist. Filamentous fungi have a more complex cytoskeleton, which may affect vesicle transport to the cell membrane [15]. Snc1 recycling requires the involvement of Drs2-Cdc50 (a phospholipid flippase), Rcy1 (an F-box protein), Snx4-Atg20 (a sorting nexin), as well as COPI (the coat protein complex (Table 1) [16], although the exact relationship between these proteins remains unclear. Studies have shown that Snc1 is crucial for protein transport in filamentous fungi, and genetic manipulations of *SNC1* result in the enhanced secretion of various enzymes such as glucose oxidase, glucoamylase, and α-amylase [17,18,19]. This review will comprehensively describe the structure and regulatory role of Snc1, its interaction with other proteins, its roles in endocytosis and exocytosis, as well as its roles in the assembly and disassembly of the SNARE complex. We will also discuss the impact of ubiquitination on the localization and recycling of Snc1, and finally analyze some strategies for enhancing protein production through the genetic manipulation of *SNC1*.

## 2. Structure of Snc1 Protein

The SNARE gene *SNC1* of *Fusarium graminearum* encodes a protein of 118 amino acids [42]. The *Trichoderma reesei SNC1*-encoded protein consists of 111 amino acids and has a predicted molecular mass of approximately 13 kDa [41]. Valkonen (2003) discovered that the Snc1 protein in *T. reesei* shares similar identity levels of 53% and 61% with the Snc1 and Snc2 proteins of *S. cerevisiae*, respectively [43]. This suggests that the Snc1 of *T. reesei* and *S. cerevisiae* may have functional similarities. Likewise, the v-SNARE paralogs Snc1 and Snc2 share a 79% identical amino acid sequence and are considered to be functionally redundant in *S. cerevisiae* [44,45]. Therefore, the *SNC* genes that encode Snc proteins in *S. cerevisiae* are thought to be duplicated. Furthermore, in *T. reesei*, the arginine at position 48 of the Snc1 protein is conserved among various R-SNAREs [43]. The synaptobrevin signature sequence is the most conserved area among these proteins. It is predicted that the putative cytoplasmic helix plays an important role during SNARE complex formation in yeast [46]. The PSIPRED server predicts that SNC1 will form two α-helices during the formation of the SNARE complex [47].

## 3. Regulatory Role of Snc1 in Endocytosis and Exocytosis

Snc1 is a protein that is highly conserved in filamentous fungi, and plays a crucial role in the maintenance of cellular homeostasis by regulating various processes such as vesicular transport, endocytosis, and exocytosis [16,19,48].

### 3.1. Exocytosis

The fusion of transport vesicles (containing exocytic cargo) with PM is tightly regulated through exocytosis. The central machineries that control the operation of exocytosis include the Rab family of GTPases, SM proteins (Sec1/Munc18) and the exocytic SNARE complex [21,49]. SM proteins regulate SNARE-mediated membrane fusion [49,50]. The exocytic complex is involved in the last steps of protein secretion, specifically the fusion of late secretory vesicles with the PM. The octameric exocyst complex arbitrates the tethering of secretory vesicles, which is followed by membrane fusion enabled by the assembly and disassembly of the SNARE complex [51]. In *S. cerevisiae*, the exocyst complex consists of the Exo70, Exo84, Sec3, Sec5, Sec6, Sec8, and Sec15 proteins [23,24]. Furthermore, Sro7 and Sro77 are two homolog proteins involved in polarized exocytosis in yeast. They interact with Snc1 to regulate its trafficking and localization to specific membrane domains [25]. Upon the arrival of secretory vesicles at the PM, the assembly of the SNARE complex is initiated, which comprises Snc1/2 (v-SNAREs localized at the secretory vesicles), and Sso1/2 and Sec9 (t-SNAREs localized at the PM) (Figure 2) [21,50]. Snc1/2 and Sso1/2 contribute one helix each, while Sec9 offers two helices for SNARE complex formation [20,21]. Some in vitro studies have revealed that a hetero-oligomeric complex of Sec9 and Sso1 binds Snc1, but not the individual t-SNARE proteins [51]. Initially, vesicles are tethered to the exocyst sites, followed by SNARE assembly, which enables the “zipping” of the membranes and subsequent vesicle fusion, leading to exocytosis [51].

Previous research endeavors have shown that Snc1 plays an essential role in establishing polarized growth and maintaining cell wall integrity in filamentous fungi by facilitating the exocytosis of cargo-containing vesicles towards hyphal tips [48]. Thus, Snc1 is essential for promoting hyphal growth and development in these organisms [19,52]. This has been suggested by Kubicek and colleagues in *T. reesei*, as they found that Sso1 and Sso2 both interact with Snc1 in the subapical and apical regions of the fungal hyphae, respectively [20]. Additionally, Snc1 regulates AmyB (α-amylase) localization at the septa and hyphal tips in *Fusarium odoratissimum* [19]. More so, Snc1 is considered crucial for the secretion of the various hydrolytic enzymes involved in nutrient acquisition and the degradation of extracellular substrates, as well as cell wall remodeling enzymes that are necessary for cell wall biosynthesis and maintenance, underscoring its role in the growth and development of fungi [16,18,53].

### 3.2. Endocytosis

Snc1 is a well-studied endocytic cargo protein that mediates the fusion of exocytic vesicles with the PM. In *S. cerevisiae*, Snc1 is recycled or internalized at the polarization sites via endocytosis [54]. The protein is incorporated into the endocytic vesicle at the initial stages of endocytosis following interaction with some endocytic proteins, such as Ede1, Syp1, Sla2, and Pal1 [34,35]. Wrasman provided valuable information regarding cargo selection, early coat protein interactions, the nucleation of the Arp2/3 complex, as well as the identification of some novel proteins participating in endocytosis [35]. Another study by Shanks et al. (2012) identified Vps41 as a critical regulator of Snc1/2 recycling. They found that Vps41 interacts with Snc1/2 to promote their sorting into recycling vesicles [33].

After the delivery of vesicles-transported cargos to the PM, the rapid internalization of SNC1 occurs and it is immediately recycled back to the endosome and Golgi membranes for another round of the trafficking event [55]. Burston et al. systematically defined the genes required for Snc1 internalization using a quantitative genome-wide screening that monitors the localization of the yeast vesicle-associated membrane protein (VAMP)/synaptobrevin, a homolog of Snc1. They placed these genes into functional modules containing known and novel endocytic regulators through genetic interaction mapping, and cargo selectivity was evaluated using an array-based comparative analysis. They demonstrated that clathrin and the yeast AP180 clathrin adaptor proteins have a cargo-specific role in Snc1 internalization [28]. Snc1 interacts with the endocytic protein Sla1 to regulate the internalization of the α-factor receptor Ste2p [29]. Meanwhile, a mutation in the sorting signal based on methionine, which is in the cytoplasmic domain of Snc1, reduces endocytosis and inhibits Snc1 recycling to Golgi from PM [36]. This mutation (Snc1-M43A) also leads to reduced growth and protein secretion in yeast, the aggregation of post-Golgi secretory vesicles, and the fragmentation of vacuoles [36]. Interestingly, cells lacking the *SNC1* gene exhibit deficiencies in the uptake of proteins from the cell surface during endocytosis [16]. The deletion of *SNC1* in *S. cerevisiae* results in a blockade in the delivery of FM4-64 to the vacuole and the endocytosis of the α-factor receptor Ste2p [56]. Xu and colleagues also found that the deletion of *SNC1* in *S. cerevisiae* results in a buildup of post-Golgi vesicles and causes secretion deficiencies [57]. Inhibitors can reduce Snc1-mediated protein secretion and alter its cellular localization. Snc1 normally localizes to the PM at the early bud stage in actively growing cells of budding yeast; however, it becomes accumulated at the Golgi after treatment with turbimycin [58]. The Snc1 ortholog in *A. nidulans*, SynA, is also a substrate for the sub-apical collar of endocytosis [59,60,61]. In *A. oryzae*, the Snc1 ortholog is predominantly observed in the Spitzenkörper, but gathers at the PM in *Aoend4* mutants, signifying defective endocytosis [62]. To sum it up, these studies demonstrate that Snc1 protein plays a major role in both anterograde and retrograde protein transport between the PM and Golgi.

SNARE complex formation is highly specific, ensuring that all components are transported to their appropriate locations through the secretory pathway and then recycled back to their respective membranes for reuse. Overall, Snc1 is essential in regulating protein secretion, maintaining cell wall integrity, managing endocytosis, and responding to nutrient availability. Its significance in these cellular processes emphasizes its central role in fungal physiology. However, the exact mechanisms through which Snc1 governs these processes are still not entirely clear.

## 4. Assembly and Disassembly of Snc1 SNARE Complex

Following membrane fusion, the v-SNARE should be retrieved back to the donor organelle or compartment, a process that requires disassembling the cis-SNARE complex; this involves the function of t-SNARE on the target membranes. A sorting signal is necessary for the concentration of proteins into endosomal vesicles, and Snc1 contains such a signal, which leads towards the endosomal pathway by targeting the respective proteins. The disassembly of the highly stable SNARE complexes necessitates ATP hydrolysis, which is facilitated by the NSF and its binding to the soluble NSF attachment protein (SNAP). Sec18 and Sec17 are, respectively, the homologs of these proteins identified in *S. cerevisiae*. It has been observed in the *sec18-1* mutant of *S. cerevisiae* that a loss of function significantly affects protein secretion at various stages [63]. Different studies have reported that these proteins (NSF/SNAP or their homologs) function on almost all the SNARE complexes [36,64,65].

There have been various theories regarding the role of Sec17/SNAP and Sec18/NSF. Originally, it was suggested that the SNARE complex disassembly initiates the membrane fusion [22]. However, Song et al. (2017) demonstrated in *sec18-1* mutant strain of *S. cerevisiae* that dissociation is not mandatory after the assembly of trans-SNARE complexes [63]. It was hypothesized that Sec18/NSF is involved in the SNARE priming step, which leads to the assembly of trans-SNARE complexes using various membranes [66]. SNAP and NSF have different binding preferences to the SNARE complex (free t-SNAREs or assembled SNAREs) [67]. This indicates that the disassembly of SNARE complexes is essential to retaining the SNAREs that are available for the composition of a new trans-SNARE complex, which requires precision in the formation of the SNARE complex.

The t-SNAREs Tlg1 and Tlg2 localize at endosomes or late-Golgi and play an essential role in Snc1 recycling from PM to Golgi [56,68]. The v-SNARE Snc1 plays a crucial role in TGN–endosome vesicle fusion and localizes both at the TGN and the PM [22]. The deletion of either Tlg1 or Tlg2 results in a loss of SNC1 localization to the PM, suggesting that vesicles derived from TGN recycle Snc1 back to the PM [22]. These findings conflict with the traditional model in which the early endosome mediates the recycling of cargos such as Snc1 back to PM independent of the TGN. A possible explanation for this discrepancy might be that Snc1 behaves as a v-SNARE for PM–endosome vesicle fusion. Later on, it is transported towards the early endosome, and subsequently trafficked to TGN, and finally recycled back to the PM. Thus, the latest pathway that emerges in yeast suggests that TGN functions as an early endosome, and Snc1 functions as a v-SNARE for PM–TGN vesicle fusion, where it undergoes a transport cycle that includes export from the PM, trafficking to the TGN, and then recycling back to the PM (Figure 2).

Ma and Burd (2019) identified two retrieval pathways from the endosomal system that are genetically distinct. The first recycling pathway is from PM, which relies on the Rcy1-F-box protein, while the second one is a retrograde pathway that originates from the prevacuole/multivesicular endosome and relies on the Snx4–Atg20 sorting nexin complex. Interestingly, the mutation of lysine residues within the transmembrane domain (TM) of Snc1 abolishes its retrograde sorting by Snx4–Atg20, causing it to be sorted towards the degradative multivesicular endosome pathway instead [26]. This study provides insights into the multiple pathways that Snc1 follows during vesicle trafficking between the endosomal system and the Golgi. Similarly, Best et al. (2020) examined the mechanisms of the post-endocytic recycling of yeast synaptobrevin, SNC1 [16]. The authors discovered that Snc1 is recycled through two distinct pathways, mediated by Rcy1-COPI (a sorting nexin), and retromer. Interestingly, the sorting of Snc1 into these two pathways is regulated by its phosphorylation status. Furthermore, the study proposed that the Rcy1-COPI-mediated recycling of Snc1 occurs mainly under conditions of enhanced membrane trafficking, and it is important for maintaining proper levels of Snc1 at the PM. Generally, the retromer-mediated recycling of Snc1 occurs under basal conditions and plays a crucial role in regulating Snc1 endosomal sorting and degradation. This study highlights the complex and regulated nature of Snc1 recycling, and the importance of different trafficking pathways involved in this process.

## 5. Impact of Ubiquitination on the Localization and Recycling of Snc1

In eukaryotic cells, chemical reactions take place in cellular compartments that are enclosed by membranes. SNARE proteins participate in the fusion of both similar (homotypic) and different (heterotypic) membranes. Typically, SNAREs are anchored in the “donor” membrane by a C-terminal transmembrane domain. The N-terminal domain consists of 60–70 amino acid hepta-peptides, known as SNARE motifs, which establish contact with a SNARE on the “recipient” membrane via a coiled-coil formation, and this results in the formation of a trans-SNARE complex. Oberhofer (2004) observed the post-translational modifications of Snc1 and Snc2 by ubiquitin in *S. cerevisiae* [30]. They found that ubiquitylation occurs on at least two lysines in the substrate and that this covalent linkage occurs through the ubiquitin ligase RSP5. In endocytic mutants, ubiquitylated Snc1 accumulates at the PM, indicating that the ubiquitylation reaction takes place at the PM. One of the ubiquitylation sites, lysine-63, is located in the coiled-coil domain of SNAREs, and its replacement with arginine result in Snc1 being mis-localized to the vacuolar membrane. The second ubiquitylation site remains unidentified. The ubiquitin binding protein DDI1 also interacts with Snc1/Snc2 and influences the availability of the SNAREs for the formation of a trans-SNARE complex. The role of the internal UBA (ubiquitin-associated) domain of DDI1 in this interaction is still unknown. Xu et al. (2017) showed that the K63 ubiquitylation of the v-SNARE Snc1 is essential for its proper recycling via COPI-coated vesicles [57]. Chen et al. (2011) also proposed that Snc1 undergoes ubiquitination to promote its recycling back to the PM [69] (Figure 3).

## 6. Various Snc1 Interacting Proteins and Their Role in Cellular Trafficking in Fungi

The *S. cerevisiae* Snc1 protein was shown to form complexes with a range of syntaxin-like t-SNAREs in vitro, including Sed5 (found in the cis-Golgi), Vam3, Vam7 (found in vacuoles), Pep12 (found in endosomes), Tlg1 and Tlg2 (found in the trans-Golgi and endosomes), Vti1, and Syn8 (Table 1) [22,26,31,32,38]. While Snc1 and Snc2 can be replaced by Sec22 and Nyv1 in vitro, these replacements cannot occur in vivo, as each protein is confined to its specific compartment [36,37]. The above findings suggest that the specificity of intracellular membrane fusion/attachment events relies on SNAREs targeting specific compartments.

Rab GTPases a play significant role in regulating various physiological processes, such as hyphal growth, organelle positioning, and cell differentiation in filamentous fungi [70]. Current studies have revealed the essential involvement of Rab GTPases in regulating the function and localization of SNC1 in filamentous fungi. For example, in *F. graminearum*, the inactivation of the Rab GTPase FgRab1 causes the blockade of Snc1 transport between the Golgi and the PM, resulting in defects during endocytic vesicle fusion with target membranes [39]. Similarly, in *Magnaporthe oryzae*, the Rab GTPase MoRab7 has been found to interact with Snc1 and regulate its transport between the TGN and the PM. The deletion of MoRab7 leads to Snc1 mis-localization in internal vesicles, resulting in impaired appressorium formation and reduced plant infection [40]. Furthermore, Fiedler et al. identified a regulatory mechanism involving the small GTPase ArfA and the phospholipid phosphatidylinositol-4-phosphate (PI4P). They discovered that ArfA interacts with Snc1 to control its localization and activity in the secretory pathway in *Aspergillus niger*, a function that is necessary for COPI coat protein recruitment to Golgi membranes. Additionally, PI4P enrichment in the Golgi and plasma membrane is required for Snc1 localization and function [27]. Genetic interaction studies have also revealed a potential role for Snc1 in nutrient availability response, as it interacts with Ras GTPase Ras2 [41].

The regulation of the Snc1 protein function Is complex and involves various molecular components, such as regulatory proteins, post-translational modifications, and interactions with other SNARE proteins. The interaction between Snc1 and other proteins is crucial for its role in protein trafficking. These findings collectively suggest that these proteins have a crucial role in regulating Snc1 trafficking in filamentous fungi; through controlling its localization and post-translational modifications, they impact physiological processes, including hyphal growth, appressorium formation, and virulence (Figure 4 and Figure 5).

## 7. Role of Snc1 in Protein Secretion and Development of Fungi

Limited research has been conducted on secretion-related genes and their expression in fungi under varying carbon source conditions. The secretion of cellulases in *T. reesei* is a highly regulated process that involves multiple proteins and pathways. One critical component of this pathway is Snc1, which plays a vital role in the transport of cellulases to the PM for exocytosis. Wu et al., (2017), in an effort to enhance the heterologous production of glucose oxidase, adopted a strategy that involved overexpressing genes implicated in the secretory pathway in *T. reesei*. They found that overexpressing *SNC1* represented a 2.2-fold increment in glucose oxidase production, as well as the increased expression of two other component genes, *HAC1* and *BIP1* [18]. The overexpression of one component gene more or less affected the expression of the other two genes, suggesting a complex regulating mechanism. This study demonstrates the potential of engineering the secretion pathway in order to enhance heterologous protein production in *T. reesei.* Similarly, Ji et al. observed an increased expression of *AnGOx* in *T. reesei* through the overexpression of *SSO2*, *SNC1*, and *RHO3*, the three important components of the secretory pathway [72].

Likewise, the over-expression of exocytic SNARE genes, including *SNC1*, resulted in a 71% increase in Cel7A’s secretion of *Talaromyces emersonii* in *S. cerevisiae* [73]. As mentioned earlier (exocytosis section), Snc1 also participates in regulating the localization of AmyB in *F. odoratissimum*. These findings suggest that Snc1 has a critical role in the secretion of cellulases and can be targeted for enhancing heterologous protein production in various organisms. In *S. cerevisiae*, the deletion of *SNC1* and *SNC2* genes results in temperature sensitivity when incubated in a synthetic medium. Moreover, the resultant strains were unable to proliferate on a nutrient-rich medium [44]. Interestingly, the overexpression of *SNC1* in *S. cerevisiae* can partially reduce the temperature sensitivity caused by the deletion of *SSO2* [46] and *SEC9* [74].

*SNC1* homologs can functionally complement each other, as shown by Valkonen (2003). They confirmed the functional homology of *T. reesei* SncI with the Snc1/2 proteins of *S. cerevisiae* by conducting complementation experiments. *T. reesei* SncI efficiently complemented both the secretion deficiency and the growth defects of *S. cerevisiae* lacking Snc1/2 proteins [43]. These findings highlight the critical role of Snc1/2 proteins in various organisms and their potential as targets for genetic engineering in order to improve growth and secretion capabilities. However, various inhibitors can affect the secretory role of Snc1. *SNC1* gene expression was affected by protein-trafficking and -folding inhibitors such as brefeldin A (BFA) and dithiothreitol (DTT) in *T. reesei*. Although they considerably reduced protein secretion, the total protein synthesis was unaffected [75]. Both inhibitors particularly effect the initial stages of the secretory pathway and inhibit protein secretion in *T. reesei* [75]. However, Gasser et al. (2008) showed that both DDT and BFA induce *SNC1* gene expression, while the induction is more considerate for BFA treatment than DDT [76]. BFA treatment resulted in a higher expression of *NSF1* and *YPT1* genes than *SNC1* [43]. BFA blocks ER-to-Golgi transport, and one possible explanation for this difference is that protein transport may be directed to vacuoles upon BFA treatment, resulting in the induced expression of *NSF1* and *YPT1*, which are involved in the initial stages of the secretory pathway. However, it showed mild effect on the final stages of the protein secretion pathway, which resulted in a lesser induction of *SNC1*. Similarly, *SNC1* expression experiences a 2–3 fold increase when the UPR (unfolded protein response) pathway is activated by DTT in *S. cerevisiae* [77]. The resultant induction may largely depend on *HAC1* and *IRE1*, which are the key components of the UPR pathway, and their deletion resulted in the reduced expression of *SNC1*. It can be proposed that the induction of *SNC1* is a secondary effect of the induction of UPR, as no putative UPR elements were found in the 5′ non-coding region of the *SNC1* gene.

Indeed, more research is required in order to completely comprehend the molecular mechanisms involved in the effect of *SNC1* overexpression on protein secretion in filamentous fungi. For instance, it will be important to decipher the role of Snc1 in the formation and trafficking of secretory vesicles, as well as its interactions with other proteins involved in the secretory pathway. In addition, it will be important to explore the potential of *SNC1* overexpression as a tool for optimizing protein secretion in biotechnological applications, such as the production of industrial enzymes and biofuels. Overall, the studies reviewed here highlight the importance of *SNC1* in regulating fungal protein secretion and the potential for its manipulation to enhance the efficiency of biotechnological processes.

## 8. Strategies for Enhancing Fungal Protein Production through *SNC1* Genetic Manipulation

Several strategies have been proposed to increase the protein production potential of filamentous fungi through *SNC1* genetic manipulation. A few of the potential strategies are as follows:Overexpression of *SNC1*: As mentioned earlier, the overexpression of *SNC1* was shown to increase fungal protein secretion. Thus, one potential strategy by which to enhance protein production would be to overexpress *SNC1* in the fungal hosts. This could be achieved through various methods, such as promoter engineering, gene modification, or increasing the gene copy number. The overexpression of *SNC1* can be achieved using strong promoters such as P*pdc1*, P*cdna1* or P*cbh1*. Genetically engineered promoters, which include the transcriptional factors of both the constitutive and inducible promoters, remain active even under different media. For more details, please visit the following article [78].The co-expression of *SNC1* along with key regulatory genes: *SNC1* is just one of many genes that play a significant role in the protein secretion of filamentous fungi. Thus, another strategy by which to enhance protein production would be to co-express *SNC1* with other genes involved in protein secretion, such as chaperones, transporters or related SNAREs (please consult [18] for reference). This could potentially enhance the efficiency and fidelity of the protein secretion pathway.Targeted deletion of *SNC1* inhibitors: Several proteins have been identified that can inhibit the function of SNC1 and other vesicular trafficking proteins. The activation of the glycogen synthase kinase (GSK-3β) retards the exocytosis of the synaptic vesicle in response to membrane depolarization [79]. Generally, the HOPS complex works synergistically with Sec17/18 during SANRE assembly and disassembly, respectively. Strikingly, HOPS inhibits the disassembly of SNARE complexes in the trans-, but not in the cis-, configuration [80]. Similarly, elevated levels of Sec17 can inhibit vacuole fusion through the recapture of primed SNAREs [81]. MED (myristoylated alanine-rich C kinase substrate effector domain) may generally inhibit vacuolar lipid rearrangements or may interfere with the essential interactions of SNAREs with lipids [81]. Pobbati et al. suggested that tomosyn can act as a negative regulator of exocytosis by inhibiting the binding of the vesicular synaptobrevin 2 to its plasma membrane acceptors [82]. Thus, a strategy by which to enhance protein production would be to genetically manipulate these inhibitors to increase the activity of Snc1.

In summary, these strategies are potentially promising for enhancing protein production in filamentous fungi through *SNC1* genetic manipulations. However, the optimization of each strategy will likely depend on the specific fungal host, target protein, and desired application. Thus, further research endeavors are required to fully explore the potential of *SNC1* genetic manipulations for biotechnological applications.

## Figures and Tables

**Figure 1 cells-12-01547-f001:**
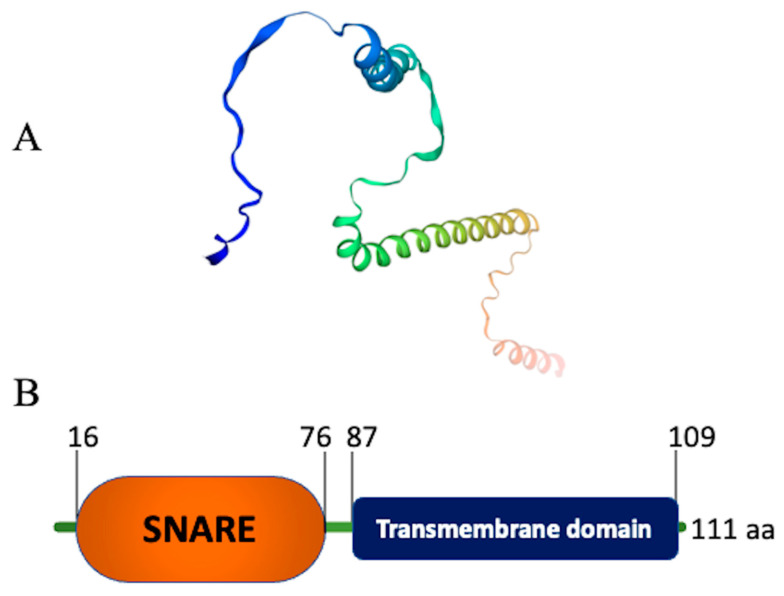
Protein structure of Snc1 of *Trichoderma reesei* QM6a. (**A**) 3D structure of SNARE protein Snc1 (created at https://swissmodel.expasy.org (accessed on 7 April 2023)). (**B**) Schematic representation of Snc1 domains (created at https://smart.embl-heidelberg.de (accessed on 7 April 2023)). SNARE domain is represented in the orange color (residues 16–76) and the trans-membrane domain is represented in the dark-blue color (residues 87–109).

**Figure 2 cells-12-01547-f002:**
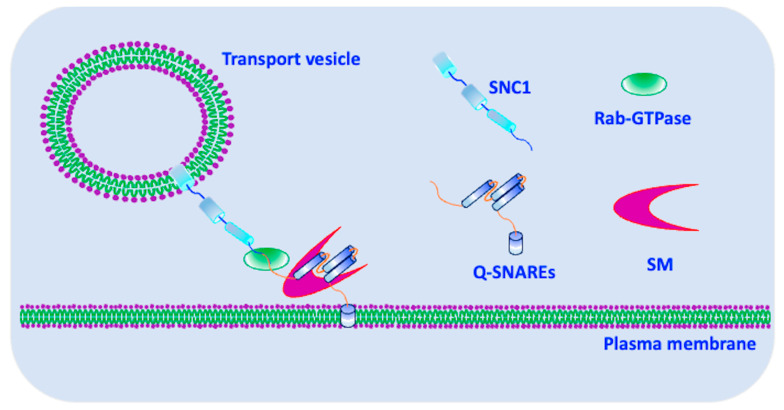
Trans-SNARE complex of the Snc1, Q-SNAREs, Rab-GTPase and SM proteins. The R-SNARE SNC1 forms a trans-SNARE complex with three Q-SNAREs (Sso1, Sso2 and Sec9). The Rab-GTPases and SM proteins (Sec1/Munc18) mediate the SNARE complex formation, as well as the docking and delivery of secretory vesicles to the plasma membrane in collaboration with SNARE proteins.

**Figure 3 cells-12-01547-f003:**
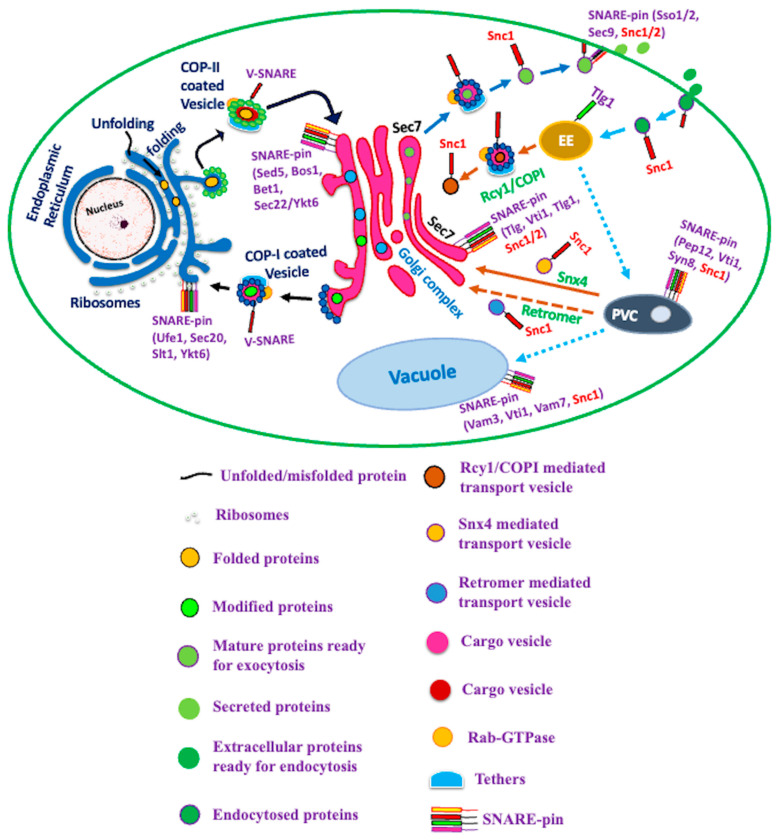
Role of SNARE protein SNC1 in vesicle trafficking and membrane fusion of fungi. SNC1 plays an essential role in anterograde and retrograde vesicle trafficking between the PM and Golgi. SNC1 interacts with exocytic SNAREs (SSo1, SSO2 and Sec9) and the exocytic complex (Sec3, Sec5, Sec6, Sec8, Sec15, Exo70, Exo84) for efficient exocytosis and releases cellular contents to the extracellular environment. The SM (Sec1/Munc18) proteins also play an important role during the exocytic fusion with the PM. During anterograde trafficking, SNC1 interacts with the endocytic proteins Tlg1, Tlg2, Sro7, Syp1, Ede1 and Pal1. The post-endocytic recycling of SNC1 is dependent upon Rcy1/COPI and Snx4 after endocytosis. The post-endocytic recycling of SNC1 adopts a minor route via the retromer. The Tlg1-positive early endosome first receives the endocytosed cargoes, which are then selectively transported by Rcy1/Drs2/COPI back to the Sec7-positive TGN or directly recycled back to the PM, or may remain in the early endosome as it matures into a PVE. Once the EE matures into a PVE, the specific cargoes are able to return by either the retromer or Snx4. The PVE will eventually fuse with the vacuole. Various other factors are also involved in the vesicular trafficking, such as Rab-GTPases, tethers and coat proteins.

**Figure 4 cells-12-01547-f004:**
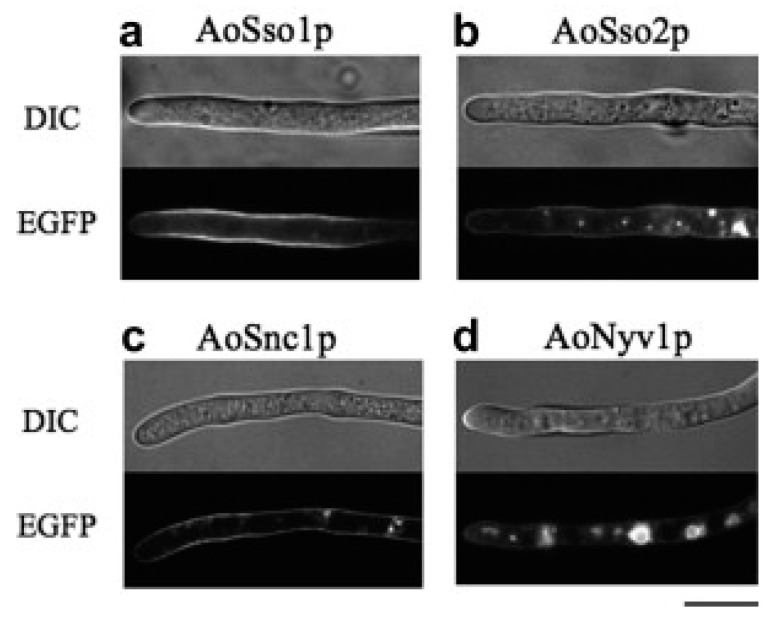
Subcellular localization of Snc1 and related SNAREs of PM in *Aspergillus oryzae*. DIC and EGFP fluorescence micrographs of strains expressing the fusion proteins of (**a**) AoSso1p, (**b**) AoSso2p, (**c**) AoSnc1p, and (**d**) AoNyv1p. The bar represents 10 µm. (Reprint from “Systematic analysis of SNARE localization in the filamentous fungus *Aspergillus oryzae*”, by Kuratsu et al., [71], Copyright (2023), by Elsevier).

**Figure 5 cells-12-01547-f005:**
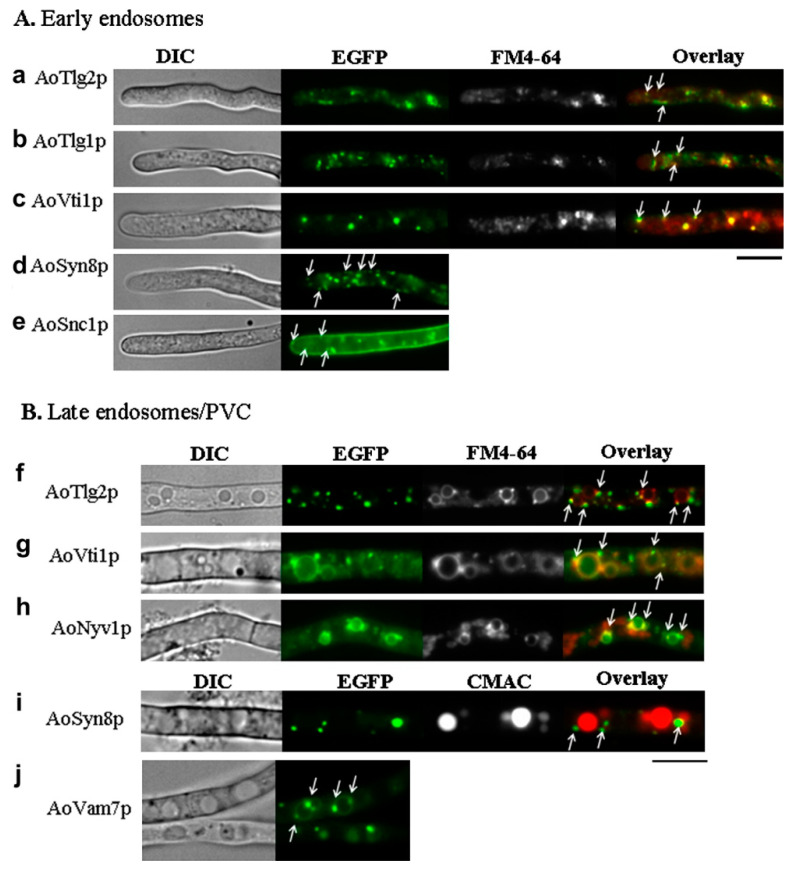
Subcellular localization of the endosome-related SNAREs of *Aspergillus oryzae*. DIC, EGFP, and FM4-64 or CMAC fluorescence micrographs of strains expressing the fusion proteins of (**A**) early endosome-resident SNAREs, including (**a**) AoTlg2p, (**b**) AoTlg1p, (**c**) AoVti1p, (**d**) AoSyn8p, and (**e**) AoSnc1p, and (**B**) late endosome-resident SNAREs, including (**f**) AoTlg2p, (**g**) AoVti1p, (**h**) AoNyv1p, (**i**) AoSyn8p, and (**j**) AoVam7p. Arrows in (**A**) indicate early endosomes that show directional movement along the hyphal axis, and arrows in (**B**) indicate late endosomes that are static and adjacent to vacuoles. The bar represents 10 µm (Reprint from “Systematic analysis of SNARE localization in the filamentous fungus *Aspergillus oryzae*”, by Kuratsu et al., [71], Copyright (2023), by Elsevier).

**Table 1 cells-12-01547-t001:** SNC1 interacting proteins of fungi.

Protein Name	Function	Interaction with SNC1	References
Sso1/Sso2	Vesicle fusion with plasma membrane	Essential for exocytosis of secretory vesicles	[20]
Sec1/Munc18	Docking of secretory vesicles and their fusion with the PM	Critical for efficient vesicle fusion with plasma membrane	[12]
Sec9	Docking of cargo vesicles and their fusion	Fusion of the secretory vesicles and PM for efficient exocytosis	[20,21]
Sec18/NSF Sec17/SNAP	Assembly and disassembly of SNAREs	Assembly and disassembly of SNC1 mediated SNAREs	[22]
Exo70/ Exo84	Important components of exocyst complex	Promotes SNC1 localization towards the exocytic sites on the PM	[23,24]
Sec3	Important components of the exocyst complex	Promote exocytic complex formation at the PM	[23,24]
Sec5			
Sec6			
Sec8			
Sec15			
Sro7/Sro77	Exocytosis and actin organization	Regulate trafficking and localization of SNC1 to specific membrane domains	[25]
Cdc42	Subunit of exocyst complex	Fusion of secretory vesicles and the PM	[10]
Drs2-Cdc50	Phospholipid flippase (Subunit of exocyst complex)	Post-endocytic recycling of SNC1	[16,26]
Rcy1	F-box protein (Subunit of exocyst complex)		
Snx4-Atg20	sorting nexin (Subunit of exocyst complex)		
COPI coat complex	COPI coat complex surrounds the cargo vesicles for cellular transportation		
ArfA	Recruitment of COPI to Golgi membranes	Regulate SNC1 localization and activity in the secretory pathway	[27]
AP180	Functional role in endocytosis	Play a cargo-specific role in SNC1 internalization	[28]
Tlg1/Tlg2	t-SNAREs localized to late Golgi and endosomes	Recycling of Snc1 protein from the PM towards Golgi	[22]
YPT1	GTPase involved in vesicle trafficking	Facilitates the fusion of vesicles	[12]
Sla1	Endocytosis and actin organization	SNC1 and Sla1 interaction regulates the internalization of the α-factor receptor Ste2	[29]
End3/End4	Endocytosis and actin organization	Efficient endocytosis of a subset of membrane proteins	[30]
Vam3/Vam7	Vesicle fusion	Efficient fusion of vesicles with the vacuole	[22,31]
Pep12	Endosomal and vacuolar trafficking	Essential for Snc1 recycling from endosomes to PM	[26,32]
Vps41	Key regulator of SNC1/2 recycling	Interacts with SNC1/2 and promotes its sorting into recycling vesicles	[33]
Ede1	Endocytic proteins	Mediate vesicle fusion during endocytosis	[34,35]
Syp1			
Pal1			
Vti1	Trans-Golgi network to endosome transportation of vesicles	Fusion of secretory vesicles and the PM	[26,32]
Sec22	ER-to-Golgi vesicular transport	Can complement SNC1 in vivo	[36,37]
Nyv1	Fusion of vesicles with the vacuole	Efficient vesicle fusion with the vacuole	[22,32]
Syn8	Endosomal t-SNARE involved in vesicle fusion	Exhibits promiscuous interactions with Snc1/2	[32]
Sed5	t-SNAREs of cis-Golgi	Mediate fusion of initial PM derived vesicles	[38]
RSP5	Ubiquitin ligase	Efficient recycling of SNC1	[30]
Rab1	Rab-GTPase Regulate vesicular trafficking	Rab1 inactivation blocks SNC1 recycling from Golgi to PM	[39]
Rab7	Rab-GTPase Regulate vesicular trafficking	Regulate SNC1 trafficking from trans-Golgi network to PM and vice versa	[40]
Ras2	Ras protein required to activate adenylate cyclase pathway	Ras2-SNC1 interaction suggest a role in response to nutrient availability	[41]

## Data Availability

Not applicable.

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
