# Peer review of "SNARE Protein Snc1 Is Essential for Vesicle Trafficking, Membrane Fusion and Protein Secretion in Fungi"

_cells, 2023, doi:10.3390/cells12111547_

Round 1
Reviewer 1 Report
The current manuscript ‘SNARE Protein SNC1 is essential for vesicle trafficking, membrane fusion and protein secretion in fungi’ by Adnan, M et al., reviews the functions and roles of the important SNARE protein, SNC1 in fungi. SNC1 protein is involved in multiple crucial cellular roles of vesicle trafficking, membrane fusion and required for the secretion of proteins in fungi. Exocytosis and endocytosis process, SNARE complex assembly and its role in membrane fusion are discussed in depth with focus on the role of SNC1 in those processes. The review discusses in detail the interactions of SNC1 with other SNARE and accessory proteins in anterograde and retrograde trafficking in cells. Towards the end the article nicely provides strategies for increasing the production and secretion of fungal proteins. Although the review is short, it provides a detailed description of the many cellular functions of SNARE protein SNC1, along with a list of its interacting partners. However, the review is lacking more figures to make it more thorough. I strongly suggest the authors to include more figures in the review to further improve it. Below are my suggestions:
1. Figure 1: Provide the actual 3D crystal/cryo-EM structure of SNC1 protein. Although the title says 'Structure of SNC1', what is shown is just a schematic of the domain organization, and not a real structure
2. The authors could include a figure of SNARE proteins and SNC1, as imaged using fluorescence microscopy in fungi, from previously published studies (from other groups) on vesicle trafficking, etc.
Author Response
We sincerely appreciate your time and effort in reviewing our manuscript entitled ‘SNARE Protein SNC1 is essential for vesicle trafficking, membrane fusion and protein secretion in fungi’. We are very thankful for your valuable suggestions, we have tried our best to address all of comments to further improve the manuscript.
Question 1. Figure 1: Provide the actual 3D crystal/cryo-EM structure of SNC1 protein. Although the title says 'Structure of SNC1', what is shown is just a schematic of the domain organization, and not a real structure
Answer: We have provided the required diagram
Figure 1. Protein structure of Snc1 of Trichoderma reesei QM6a A) 3D structure of SNARE protein Snc1 (created at https://swissmodel.expasy.org) B) Schematic representation of Snc1 domains (created at https://smart.embl-heidelberg.de). SNARE domain is represented in orange color (residues 16-76) and trans-membrane domain is represented in dark blue color (residues 87-109).
Question 2. The authors could include a figure of SNARE proteins and SNC1, as imaged using fluorescence microscopy in fungi, from previously published studies (from other groups) on vesicle trafficking, etc.
Answer: We have provided the required diagrams
Figure 4. Subcellular localization of SNC1 and related SNAREs of PM in Aspergillus oryzae. DIC and EGFP fluorescence micrographs of strains expressing the fusion proteins of (a) AoSso1p, (b) AoSso2p, (c) AoSnc1p, and (d) AoNyv1p. The bar represents 10 µm. (Reprint from “Systematic analysis of SNARE localization in the filamentous fungus Aspergillus oryzae”, by Kuratsu et al., 2007, Fungal Genetics and Biology, 44(12), pp.1310-1323. Copyright (2023), by Elsevier).
Figure 5. Subcellular localization of the endosome related SNAREs of Aspergillus oryzae. DIC, EGFP, and FM4-64 or CMAC fluorescence micrographs of strains expressing the fusion proteins of (A) early endosome-resident SNAREs including (a) AoTlg2p, (b) AoTlg1p, (c) AoVti1p, (d) AoSyn8p, and (e) AoSnc1p, and (B) late endosome-resident SNAREs including (f) AoTlg2p, (g) AoVti1p, (h) AoNyv1p, (i) AoSyn8p, and (j) AoVam7p are shown. Arrows in (A) indicate early endosomes that show directional movement along the hyphal axis, and arrows in (B) indicate late endosomes that are static and adjacent to vacuoles. The bar represents 10 µm. (Reprint from “Systematic analysis of SNARE localization in the filamentous fungus Aspergillus oryzae”, by Kuratsu et al., 2007, Fungal Genetics and Biology, 44(12), pp.1310-1323. Copyright (2023), by Elsevier).

Reviewer 2 Report
Adnan et al. wrote a review on SNC1. Overall, my impression is that besides its well characterized role as a SNARE protein, I was not able to gain a new insight on its function after reading this manuscript. It would have been useful to me if more emphasis was given to its unique roles in fungal settings.
1. Introduction
The authors initially used ‘v-’ and ‘t-SNARE’ terminology and then suddenly used ‘R-’ and ‘Q-SNARE’ terminology.
3. Regulator role of SNC1
The first introductory paragraph doesn’t have a purpose to the ensuing body of text.
I think a figure of trans SNARE complex involving SNC1 and other SNARES including SM and Rab would be useful.
3.2. Endocytosis
Line 160: “SNC1 is cycled…”. Does it mean “SNC1 is internalized”?
Line 170 – 173: This sentence is very confusing.
4. Assembly and disassembly
Line 222-223: This sentence is grammatically wrong.
Line 227: ‘traditional’ to ‘conventional’.
6. Role of SNC1 in protein secretion.
I think the almost all of the first paragraph is unrelated to the theme of this section.
Line 349-350: confusing sentence.
Line 352: ‘secretorion’ to ‘secretory’.
Line 353-354: I cannot understand the purpose of this sentence.
Line 357: “As mentioned earlier, …..”. Where?
Line 361: “in a synthetic medium, moreover” to “in a synthetic medium. Moreover”.
Line 378-381: As BFA blocks ER-to-Golgi transport, the authors should interpret the data with this information.
7. Strategies
Page 11: It would be nice to include examples for targeted deletion of snc1 inhibitors.
Line 418-422: This paragraph does not add to the theme of section 7.
It would help if this manuscript is edited by a native English speaker.
Author Response
Introduction:
Question 1:
The authors initially used ‘v-’ and ‘t-SNARE’ terminology and then suddenly used ‘R-’ and ‘Q-SNARE’ terminology.
Answer: SNAREs are named after their specific residues, arginine (R) or glutamine (Q). A SNARE complex involves three Q-SNAREs and one R-SNARE. This classification is very important in terms of SNARE interaction and complex formation. However, the subcellular localization of a SNARE protein on transport vesicle or target membrane (plasma membrane, ER membrane, Golgi membrane etc.) helps in their identification as v-SNARE or t-SNARE, respectively. Both of these terminologies are very important to differentiate their subcellular localization and specific roles during SNARE interaction.
- Regulatory role of SNC1
Question 2: The first introductory paragraph doesn’t have a purpose to the ensuing body of text.
Answer: We have modified this part
Question 3:
I think a figure of trans SNARE complex involving SNC1 and other SNARES including SM and Rab would be useful.
Answer: We have provided the required diagram
Figure 2. Trans-SNARE complex of Snc1, Q-SNAREs, Rab-GTPase and SM proteins. The R-SNARE Snc1 forms trans-SNARE complex with three Q-SNAREs (Sso1, Sso2 and SEC9). Rab-GTPases and SM proteins (Sec1/Munc18) mediate the SNARE complex formation as well as docking and delivery of secretory vesicles to plasma membrane in collaboration with SNARE proteins.
3.2. Endocytosis
Question 4:
Line 160: “SNC1 is cycled…”. Does it mean “SNC1 is internalized”?
Answer: We have a modified this sentence as “In S. cerevisiae, SNC1 is recycled or internalized at the polarization sites via endocytosis”
Question 5:
Line 170 – 173: This sentence is very confusing.
Answer: We have modified the sentence as
“Burston et al. had systematically defined genes required for internalization using a quantitative genome-wide screen that monitors localization of the yeast vesicle-associated membrane protein (VAMP)/synaptobrevin homologue Snc1. They placed these genes into functional modules containing known and novel endocytic regulators through genetic interaction mapping, and cargo selectivity was evaluated by an array-based comparative analysis. They demonstrated that clathrin and the yeast AP180 clathrin adaptor proteins have a cargo-specific role in Snc1 internalization”.
- Assembly and disassembly
Question 6:
Line 222-223: This sentence is grammatically wrong.
Answer: We have removed the grammatical errors.
The t-SNAREs Tlg1 and Tlg2 localize at endosomes or late-Golgi and play essential role in Snc1 recycling from PM to Golgi
Question 7:
Line 227: ‘traditional’ to ‘conventional’.
Answer: We have modified this sentence.
“These findings conflict with the traditional model in which the early endosome mediates the recycling of cargoes such as Snc1 back to the PM independently of the TGN”.
- Role of SNC1 in protein secretion.
Question 8:
I think the almost all of the first paragraph is unrelated to the theme of this section.
Answer: We have removed the irrelevant part of this paragraph.
Question 9:
Line 349-350: confusing sentence.
Answer: We have modified the sentence, here is the response:
The overexpression of one component gene more or less affected the expression of the other two genes, suggesting a complex regulating mechanism. This study demonstrates the potential of engineering the secretion pathway for enhancing heterologous protein production in T. reesei.
Question 10:
Line 352: ‘secretorion’ to ‘secretory’.
Answer: We have modified this word accordingly.
Question 11:
Line 353-354: I cannot understand the purpose of this sentence.
Answer: We have deleted this sentence
Question 12:
Line 357: “As mentioned earlier, …..”. Where?
Answer: We have mentioned it earlier, please visit “Snc1 regulates the AmyB (α-amylase) localization at septa and hyphal tips in Fusarium odoratissimum”. We have further modified the sentence for convenience as “As mentioned earlier (exocytosis section), Snc1 also participates in regulating the localization of AmyB in F. odoratissimum”.
Question 13:
Line 361: “in a synthetic medium, moreover” to “in a synthetic medium. Moreover”.
Answer: We appreciate your valuable suggestions. We have modified this sentence accordingly.
Question 14:
Line 378-381: As BFA blocks ER-to-Golgi transport, the authors should interpret the data with this information.
Answer:
We have modified this part as follows
“As, BFA blocks ER-to-Golgi transport, thus one possible explanation for this difference is that protein transport may be directed to vacuoles upon BFA treatment, resulting in the induced expression of nsf1 and ypt1 which are involved in the initial stages of the secretory pathway. However, it showed mild effect on the final stages of protein secretion pathway, which resulted in a lesser induction of snc1. Similarly, snc1 expression is 2-3 fold enhanced, when the UPR (unfolded protein response) pathway is activated by DTT in S. cerevisiae [80]. The resultant induction may largely depend on hac1 and ire1 which are the key components of UPR pathway; and their deletion resulted in reduced expression of snc1. It can be suggested that the induction of snc1 is a secondary effect of the UPR induction, as no putative UPR-elements are found in the 5' non-coding region of snc1 gene”.
- Strategies
Question 15:
Page 11: It would be nice to include examples for targeted deletion of SNC1 inhibitors.
Answer: We have included some examples of targeted deletion of SNC1 inhibitors. Which are as follows:
Targeted deletion of SNC1 inhibitors: Several proteins have been identified that can inhibit the function of SNC1 and other vesicular trafficking proteins. Activation of the glycogen synthase kinase (GSK-3β) retards the synaptic vesicle exocytosis in response to membrane depolarization [78]. Generally, HOPS complex works synergistically with Sec17/18 during SANRE assembly and disassembly, respectively. Strikingly, HOPS inhibits the disassembly of SNARE complexes in the trans-, but not in the cis-, configuration [79]. Similarly, elevated levels of Sec17 can inhibit vacuole fusion through recapture of primed SNAREs [80]. MED (myristoylated alanine-rich C kinase substrate effector domain) may generally inhibit vacuolar lipid rearrangements or may interfere with essential interactions of SNAREs with lipids [80]. Pobbati et al., suggested that tomosyn can act as a negative regulator of exocytosis by inhibiting binding of the vesicular synaptobrevin 2 to its plasma membrane acceptors [81]. Thus, strategy to enhance protein production would be to genetically manipulate these inhibitors to increase the activity of SNC1.
Question 16:
Line 418-422: This paragraph does not add to the theme of section 7.
Answer: We have deleted this paragraph
Comments on the Quality of English Language
Question 17:
It would help if this manuscript is edited by a native English speaker.
Answer: This article has been edited by an expert to address the grammatical errors and further improved the readablilty.
*Note: There is some problem in line numbering, which is not continuous after applying the track changes. Moreover, we have added the references using Zotero software, and the references added during revision cannot be tracked due to malfunction of the software. However, we mention here that the references from 78-81 are newly added.

Round 2
Reviewer 2 Report
Significant revision was made.
It still needs minor editing.